# THREE MECHANISMS OF WEIGHT DECAY REGULARIZATION

**Guodong Zhang, Chaoqi Wang, Bowen Xu, Roger Grosse**
University of Toronto, Vector Institute
{gdzhang, cqwang, bowenxu, rgrosse}@cs.toronto.edu

## ABSTRACT

Weight decay is one of the standard tricks in the neural network toolbox, but the reasons for its regularization effect are poorly understood, and recent results have cast doubt on the traditional interpretation in terms of $L_2$ regularization. Literal weight decay has been shown to outperform $L_2$ regularization for optimizers for which they differ. We empirically investigate weight decay for three optimization algorithms (SGD, Adam, and K-FAC) and a variety of network architectures. We identify three distinct mechanisms by which weight decay exerts a regularization effect, depending on the particular optimization algorithm and architecture: (1) increasing the effective learning rate, (2) approximately regularizing the input-output Jacobian norm, and (3) reducing the effective damping coefficient for second-order optimization. Our results provide insight into how to improve the regularization of neural networks.

## 1   INTRODUCTION

Weight decay has long been a standard trick to improve the generalization performance of neural networks (Krogh & Hertz, 1992; Bos & Chug, 1996) by encouraging the weights to be small in magnitude. It is widely interpreted as a form of $L_2$ regularization because it can be derived from the gradient of the $L_2$ norm of the weights in the gradient descent setting. However, several findings cast doubt on this interpretation:

- Weight decay has sometimes been observed to improve training accuracy, not just generalization performance (e.g. Krizhevsky et al. (2012)).

- Loshchilov & Hutter (2017) found that when using Adam (Kingma & Ba, 2014) as the optimizer, literally applying weight decay (i.e. scaling the weights by a factor less than 1 in each iteration) enabled far better generalization than adding an $L_2$ regularizer to the training objective.

- Weight decay is widely used in networks with Batch Normalization (BN) (Ioffe & Szegedy, 2015). In principle, weight decay regularization should have no effect in this case, since one can scale the weights by a small factor without changing the network's predictions. Hence, it does not meaningfully constrain the network's capacity.

The effect of weight decay remains poorly understood, and we lack clear guidelines for which tasks and architectures it is likely to help or hurt. A better understanding of the role of weight decay would help us design more efficient and robust neural network architectures.

In order to better understand the effect of weight decay, we experimented with both weight decay and $L_2$ regularization applied to image classifiers using three different optimization algorithms: SGD, Adam, and Kronecker-Factored Approximate Curvature (K-FAC) (Martens & Grosse, 2015). Consistent with the observations of Loshchilov & Hutter (2017), we found that weight decay consistently outperformed $L_2$ regularization in cases where they differ. Weight decay gave an especially strong performance boost to the K-FAC optimizer, and closed most of the generalization gaps between first- and second-order optimizers, as well as between small and large batches. We then investigated the reasons for weight decay's performance boost. Surprisingly, we identified three distinct mechanisms by which weight decay has a regularizing effect, depending on the particular algorithm and architecture:

**Figure 1:** Comparison of test accuracy of the networks trained with different optimizers on both CIFAR10 and CIFAR100. We compare **Weight Decay** regularization to $L_2$ regularization and the **Baseline** (which used neither). Here, **BN+Aug** denotes the use of BN and data augmentation. **K-FAC-G** and **K-FAC-F** denote K-FAC using Gauss-Newton and Fisher matrices as the preconditioner, respectively. The results suggest that weight decay leads to improved performance across different optimizers and settings.

1. In our experiments with first-order optimization methods (SGD and Adam) on networks with BN, we found that it acts by way of the effective learning rate. Specifically, weight decay reduces the scale of the weights, increasing the effective learning rate, thereby increasing the regularization effect of gradient noise (Neelakantan et al., 2015; Keskar et al., 2016). As evidence, we found that almost all of the regularization effect of weight decay was due to applying it to layers with BN (for which weight decay is meaningless). Furthermore, when we computed the effective learning rate for the network with weight decay, and applied the same effective learning rate to a network without weight decay, this captured the full regularization effect.

2. We show that when K-FAC is applied to a linear network using the Gauss-Newton metric (K-FAC-G), weight decay is equivalent to regularizing the squared Frobenius norm of the input-output Jacobian (which was shown by Novak et al. (2018) to improve generalization). Empirically, we found that even for (nonlinear) classification networks, the Gauss-Newton norm (which K-FAC with weight decay is implicitly regularizing) is highly correlated with the Jacobian norm, and that K-FAC with weight decay significantly reduces the Jacobian norm.

3. Because the idealized, undamped version of K-FAC is invariant to affine reparameterizations, the implicit learning rate effect described above should not apply. However, in practice the approximate curvature matrix is damped by adding a multiple of the identity matrix, and this damping is not scale-invariant. We show that without weight decay, the weights grow large, causing the effective damping term to increase. If the effective damping term grows large enough to dominate the curvature term, it effectively turns K-FAC into a first-order optimizer. Weight decay keeps the effective damping term small, enabling K-FAC to retain its second-order properties, and hence improving generalization.

Hence, we have identified three distinct mechanisms by which weight decay improves generalization, depending on the optimization algorithm and network architecture. Our results underscore the subtlety and complexity of neural network training: the final performance numbers obscure a variety of complex interactions between phenomena. While more analysis and experimentation is needed to understand how broadly each of our three mechanisms applies (and to find additional mechanisms!), our work provides a starting point for understanding practical regularization effects in neural network training.

## 2 PRELIMINARIES

**Supervised learning.** Given a training set $\mathcal{S}$ consisting of training pairs $\{\mathbf{x}, y\}$, and a neural network $f_{\boldsymbol{\theta}}(\mathbf{x})$ with parameters $\boldsymbol{\theta}$ (including weights and biases), our goal is to minimize the emprical risk expressed as an average of a loss $\ell$ over the training set: $\mathcal{L}(\boldsymbol{\theta}) \equiv \frac{1}{N} \sum_{(\mathbf{x},y)\sim\mathcal{S}} \ell\left(y, f_{\boldsymbol{\theta}}(\mathbf{x})\right)$.

**Stochastic Gradient Descent.** To minimize the empirical risk $\mathcal{L}(\boldsymbol{\theta})$, stochastic gradient descent (SGD) is used extensively in deep learning community. Typically, gradient descent methods can be derived from the framework of steepest descent with respect to standard Euclidean metric in parameter space. Specifically, gradient descent minimizes the following surrogate objective in each iteration:

$$h(\boldsymbol{\theta}) = \Delta\boldsymbol{\theta}^\top \nabla_{\boldsymbol{\theta}}\mathcal{L}(\boldsymbol{\theta}) + 1/\eta \mathrm{D}(\boldsymbol{\theta}, \boldsymbol{\theta} + \Delta\boldsymbol{\theta}), \tag{1}$$

where the distance (or dissimilarity) function $\mathrm{D}(\boldsymbol{\theta}, \boldsymbol{\theta} + \Delta\boldsymbol{\theta})$ is chosen as $\frac{1}{2}\|\Delta\boldsymbol{\theta}\|_2^2$. In this case, solving equation 1 yields $\Delta\boldsymbol{\theta} = -\eta\nabla_{\boldsymbol{\theta}}\mathcal{L}(\boldsymbol{\theta})$, where $\eta$ is the learning rate.

**Natural gradient.** Though popular, gradient descent methods often struggle to navigate "valleys" in the loss surface with ill-conditioned curvature (Martens, 2010). Natural gradient descent, as a variant of second-order methods (Martens, 2014), is able to make more progress per iteration by taking into account the curvature information. One way to motivate natural gradient descent is to show that it can be derived by adapting steepest descent formulation, much like gradient descnet, except using an alternative local distance. The distance function which leads to natural gradient is the KL divergence on the model's predictive distribution $D_{KL}(p_{\boldsymbol{\theta}} \| p_{\boldsymbol{\theta}+\Delta\boldsymbol{\theta}}) \approx \frac{1}{2}\Delta\boldsymbol{\theta}^{\top}\mathbf{F}\Delta\boldsymbol{\theta}$, where $\mathbf{F}(\boldsymbol{\theta})$ is the Fisher information matrix[1] (Amari, 1998):

$$\mathbf{F} = \mathbb{E}\left[\nabla_{\boldsymbol{\theta}} \log p(y|\mathbf{x}, \boldsymbol{\theta})\nabla_{\boldsymbol{\theta}} \log p(y|\mathbf{x}, \boldsymbol{\theta})^{\top}\right]. \tag{2}$$

Applying this distance function to equation 1, we have $\boldsymbol{\theta}^{t+1} \leftarrow \boldsymbol{\theta}^t - \eta\mathbf{F}^{-1}\nabla_{\boldsymbol{\theta}}\mathcal{L}(\boldsymbol{\theta})$.

**Gauss-Newton algorithm.** Another sensible distance function in equation 1 is the $L_2$ distance on the output (logits) of the neural network, i.e. $\frac{1}{2}\|f_{\boldsymbol{\theta}+\Delta\boldsymbol{\theta}} - f_{\boldsymbol{\theta}}\|_2^2$. This leads to the *classical* Gauss-Newton algorithm which updates the parameters by $\boldsymbol{\theta}^{t+1} \leftarrow \boldsymbol{\theta}^t - \eta\mathbf{G}^{-1}\nabla_{\boldsymbol{\theta}}\mathcal{L}(\boldsymbol{\theta})$, where the Gauss-Newton (GN) matrix is defined as

$$\mathbf{G} = \mathbb{E}\left[\mathbf{J}_{\boldsymbol{\theta}}^{\top}\mathbf{J}_{\boldsymbol{\theta}}\right], \tag{3}$$

and $\mathbf{J}_{\boldsymbol{\theta}}$ is the Jacobian of $f_{\boldsymbol{\theta}}(\mathbf{x})$ w.r.t $\boldsymbol{\theta}$. The Gauss-Newton algorithm, much like natural gradient descent, is also invariant to the specific parameterization of neural network function $f_{\boldsymbol{\theta}}$.

**Two curvature matrices.** It has been shown that the GN matrix is equivalent to the Fisher matrix in the case of regression task with squared error loss (Heskes, 2000). However, they are not identical for the case of classification, where cross-entropy loss is commonly used. Nevertheless, Martens (2014) showed that the Fisher matrix is equivalent to *generalized* GN matrix when model prediction $p(y|\mathbf{x}, \boldsymbol{\theta})$ corresponds to exponential family model with natural parameters given by $f_{\boldsymbol{\theta}}(\mathbf{x})$, where the *generalized* GN matrix is given by

$$\mathbf{G} = \mathbb{E}\left[\mathbf{J}_{\boldsymbol{\theta}}^{\top}\mathbf{H}_{\ell}\mathbf{J}_{\boldsymbol{\theta}}\right], \tag{4}$$

and $\mathbf{H}_{\ell}$ is the Hessian of $\ell(y, z)$ w.r.t $z$, evaluated at $z = f_{\boldsymbol{\theta}}(\mathbf{x})$. In regression with squared error loss, the Hessian $\mathbf{H}_{\ell}$ happens to be identity matrix.

**Preconditioned gradient descent.** Given the fact that both natural gradient descent and Gauss-Newton algorithm precondition the gradient with an extra curvature matrix $\mathbf{C}(\boldsymbol{\theta})$ (including the Fisher matrix and GN matrix), we also term them *preconditioned gradient descent* for convenience.

**K-FAC.** As modern neural networks may contain millions of parameters, computing and storing the exact curvature matrix and its inverse is impractical. Kronecker-factored approximate curvature (K-FAC) (Martens & Grosse, 2015) uses a Kronecker-factored approximation to the curvature matrix to perform efficient approximate natural gradient updates. As shown by Luk & Grosse (2018), K-FAC can be applied to general pullback metric, including Fisher metric and the Gauss-Newton metric. For more details, we refer reader to Appendix F or Martens & Grosse (2015).

**Batch Normalization.** Broadly speaking, Batch Normalization (BN) is a mechanism that aims to stabilize the distribution (over a mini-batch) of inputs to a given network layer during training. This is achieved by augmenting the network with additional layers that subtract the mean $\mu$ and divide by the standard deviation $\sigma$. Typically, the normalized inputs are also scaled and shifted based on trainable parameters $\gamma$ and $\beta$:

$$\text{BN}(\mathbf{x}) = \frac{\mathbf{x} - \mu}{\sigma} \cdot \gamma + \beta. \tag{5}$$

For clarity, we ignore the parameters $\gamma$ and $\beta$, which do not impact the performance in practice. This is not surprising, since with ReLU activations, only the $\gamma$ of the last layer affects network's outputs which can be merged with the softmax layer weights (as also pointed out by van Laarhoven (2017)).

## 3 THE EFFECTIVENESS OF WEIGHT DECAY

Our goal is to understand weight decay regularization in the context of training deep neural networks. Towards this, we first discuss the relationship between $L_2$ regularization and weight decay in different optimizers.

---

[1]The underlying distribution in equation 2 has been left ambiguous. Throughout the experiments, we sample the targets from the model's predictions, as done in Martens & Grosse (2015); Zhang et al. (2017)

**Table 1:** Classification results on CIFAR-10 and CIFAR-100. **B** denotes BN while **D** denotes data augmentation, including horizontal flip and random crop. **WD** denotes weight decay regularization. Weight decay regularization improves the generalization consistently. Interestingly, we observe that weight decay gives an especially strong performance boost to the K-FAC optimizer when BN is turned off.

| Dataset | Network | B | D | SGD | | ADAM | | K-FAC-F | | K-FAC-G | |
|---------|---------|---|---|-----|-----|------|-----|---------|-----|---------|-----|
| | | | | | WD | | WD | | WD | | WD |
| CIFAR-10 | VGG16 | | | 83.20 | 84.87 | 83.16 | 84.12 | 85.58 | 89.60 | 83.85 | **89.81** |
| | | ✓ | | 86.99 | 88.85 | 88.45 | 88.72 | 87.97 | 89.02 | 88.17 | **89.77** |
| | | ✓ | ✓ | 91.71 | 93.39 | 92.89 | 93.62 | 93.12 | **93.90** | 93.19 | 93.80 |
| CIFAR-10 | ResNet32 | | | 85.47 | 86.63 | 84.43 | 87.54 | 86.82 | 90.22 | 85.24 | **90.64** |
| | | ✓ | | 86.13 | 90.65 | 89.46 | 90.61 | 89.78 | **91.24** | 89.94 | 90.91 |
| | | ✓ | ✓ | 92.95 | 95.14 | 93.63 | 94.66 | 93.80 | **95.35** | 93.44 | 95.04 |
| CIFAR-100 | VGG16 | ✓ | ✓ | 68.42 | 73.31 | 69.88 | **74.22** | 71.05 | 73.36 | 67.46 | 73.57 |
| CIFAR-100 | ResNet32 | ✓ | ✓ | 73.61 | 77.73 | 73.60 | 77.40 | 74.49 | 78.01 | 73.70 | **78.02** |

Gradient descent with weight decay is defined by the following update rule: $\boldsymbol{\theta}^{t+1} \leftarrow (1 - \eta\beta)\boldsymbol{\theta}^t - \eta\nabla\mathcal{L}(\boldsymbol{\theta}^t)$, where $\beta$ defines the rate of the weight decay per step and $\eta$ is the learning rate. In this case, weight decay is equivalent to $L_2$ regularization. However, the two differ when the gradient update is preconditioned by a matrix $\mathbf{C}^{-1}$, as in Adam or K-FAC. The preconditioned gradient descent update with $L_2$ regularization is given by

$$\boldsymbol{\theta}^{t+1} \leftarrow (\mathbf{I} - \eta\beta\mathbf{C}^{-1})\boldsymbol{\theta}^t - \eta\mathbf{C}^{-1}\nabla_{\boldsymbol{\theta}}\mathcal{L}(\boldsymbol{\theta}^t), \tag{6}$$

whereas the weight decay update is given by

$$\boldsymbol{\theta}^{t+1} \leftarrow (1 - \eta\beta)\boldsymbol{\theta}^t - \eta\mathbf{C}^{-1}\nabla_{\boldsymbol{\theta}}\mathcal{L}(\boldsymbol{\theta}^t). \tag{7}$$

The difference between these updates is whether the preconditioner is applied to $\boldsymbol{\theta}^t$. The latter update can be interpreted as the preconditioned gradient descent update on a regularized objective where the regularizer is the squared $\mathbf{C}$-norm $\|\boldsymbol{\theta}\|_{\mathbf{C}}^2 = \boldsymbol{\theta}^\top\mathbf{C}\boldsymbol{\theta}$. If $\mathbf{C}$ is adapted based on statistics collected during training, as in Adam or K-FAC, this interpretation holds only approximately because gradient descent on $\|\boldsymbol{\theta}\|_{\mathbf{C}}^2$ would require differentiating through $\mathbf{C}$. However, this approximate regularization term can still yield insight into the behavior of weight decay. (As we discuss later, this observation informs some, but not all, of the empirical phenomena we have observed.) Though the difference between the two updates may appear subtle, we find that it makes a substantial difference in terms of generalization performance.

**Initial Experiments.** We now present some empirical findings about the effectiveness of weight decay which the rest of the paper is devoted to explaining. Our experiments were carried out on two different datasets: CIFAR-10 and CIFAR-100 (Krizhevsky & Hinton, 2009) with varied batch sizes. We test VGG16 (Simonyan & Zisserman, 2014) and ResNet32 (He et al., 2016) on both CIFAR-10 and CIFAR-100 (for more details, see Appendix A). In particular, we investigate three different optimization algorithms: SGD, Adam and K-FAC. We consider two versions of K-FAC, which use the Gauss-Newton matrix (K-FAC-G) and Fisher information matrix (K-FAC-F).

Figure 1 shows the comparison between weight decay, $L_2$ regularization and the baseline. We also compare weight decay to the baseline on more settings and report the final test accuracies in Table 1. Finally, the results for large-batch training are summarized in Table 3. Based on these results, we make the following observations regarding weight decay:

1. In all experiments, weight decay regularization consistently improved the performance and was more effective than $L_2$ regularization in cases where they differ (See Figure 1).

2. Weight decay closed most of the generalization gaps between first- and second-order optimizers, as well as between small and large batches (See Table 1 and Table 3).

3. Weight decay significantly improved performance even for BN networks (See Table 1), where it does not meaningfully constrain the networks' capacity.

4. Finally, we notice that weight decay gave an especially strong performance boost to the K-FAC optimizer when BN was disabled (see the first and fourth rows in Table 1).

In the following section, we seek to explain these phenomena. With further testing, we find that weight decay can work in unexpected ways, especially in the presence of BN.

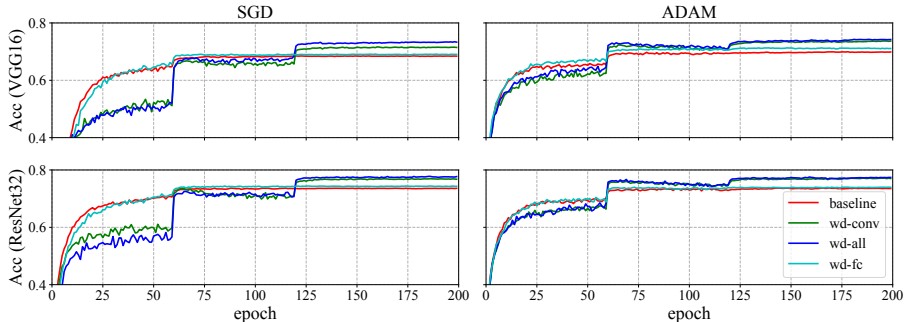

**Figure 2:** Test accuracy as a function of training epoch for SGD and Adam on CIFAR-100 with different weight decay regularization schemes. baseline is the model without weight decay; wd-conv is the model with weight decay applied to all convolutional layers; wd-all is the model with weight decay applied to all layers; wd-fc is the model with weight decay applied to the last layer (fc). Most of the generalization effect of weight decay is due to applying it to layers with BN.

## 4 THREE MECHANISMS OF WEIGHT DECAY REGULARIZATION

### 4.1 MECHANISM I: HIGHER EFFECTIVE LEARNING RATE

As discussed in Section 3, when SGD is used as the optimizer, weight decay can be interpreted as penalizing the $L_2$ norm of the weights. Classically, this was believed to constrain the model by penalizing explanations with large weight norm. However, for a network with Batch Normalization (BN), an $L_2$ penalty does not meaningfully constrain the reprsentation, because the network's predictions are invariant to rescaling of the weights and biases. More precisely, if $\text{BN}(\mathbf{x}; \boldsymbol{\theta}_l)$ denotes the output of a layer with parameters $\boldsymbol{\theta}_l$ in which BN is applied before the activation function, then

$$\text{BN}(\mathbf{x}; \alpha\boldsymbol{\theta}_l) = \text{BN}(\mathbf{x}; \boldsymbol{\theta}_l), \tag{8}$$

for any $\alpha > 0$. By choosing small $\alpha$, one can make the $L_2$ norm arbitrarily small without changing the function computed by the network. Hence, in principle, adding weight decay to layers with BN should have no effect on the optimal solution. But empirically, weight decay appears to significantly improve generalization for BN networks (e.g. see Figure 1).

van Laarhoven (2017) observed that $L_2$ regularization has an influence on the effective learning rate in (stochastic) gradient descent. In this work, we extend this result to first-order optimizers (including SGD and Adam) that weight decay increases the effective learning rate by reducing the scale of the weights. Since higher learning rates lead to larger gradient noise, which has been shown to act as a stochastic regularizer (Neelakantan et al., 2015; Keskar et al., 2016; Jastrzębski et al., 2017; Hoffer et al., 2017), this means weight decay can indirectly exert a regularizing effect through the effective learning rate. In this section, we provide additional evidence supporting the hypothesis of van Laarhoven (2017). For simplicity, this section focuses on SGD, but we've observed similar behavior when Adam is used as the optimizer.

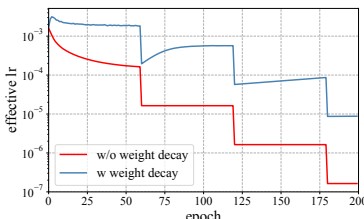

**Figure 3:** Effective learning rate of the first layer of ResNet32 trained with SGD on CIFAR-100. Without weight decay regularization, the effective learning rate decreases quickly in the beginning.

Due to its invariance to the scaling of the weights, the key property of the weight vector is its direction. As shown by Hoffer et al. (2018), the weight direction $\hat{\boldsymbol{\theta}}_l = \boldsymbol{\theta}_l/\|\boldsymbol{\theta}_l\|_2$ is updated according to

$$\hat{\boldsymbol{\theta}}_l^{t+1} \leftarrow \hat{\boldsymbol{\theta}}_l^t - \eta\|\boldsymbol{\theta}_l^t\|_2^{-2}(\mathbf{I} - \hat{\boldsymbol{\theta}}_l^t \hat{\boldsymbol{\theta}}_l^{t\top})\nabla_{\boldsymbol{\theta}_l}\mathcal{L}(\hat{\boldsymbol{\theta}}^t) + O(\eta^2). \tag{9}$$

Therefore, the effective learning rate is approximately proportional to $\eta/\|\boldsymbol{\theta}_l\|_2^2$. Which means that by decreasing the scale of the weights, weight decay regularization increases the effective learning rate.

Figure 3 shows the effective learning rate over time for two BN networks trained with SGD (the results for Adam are similar), one with weight decay and one without it. Each network is trained with a typical learning rate decay schedule, including 3 factor-of-10 reductions in the learning rate parameter, spaced 60 epochs apart. Without weight decay, the normalization effects cause an additional effective learning rate decay (due to the increase of weight norm), which reduces the effective learning rate by a factor

of 10 over the first 50 epochs. By contrast, when weight decay is applied, the effective learning rate remains more or less constant in each stage.

We now show that the effective learning rate schedule explains nearly the entire generalization effect of weight decay. First, we independently varied whether weight decay was applied to the top layer of the network, and to the remaining layers. Since all layers except the top one used BN, it's only in the top layer that weight decay would constrain the model. Training curves for SGD and Adam under all four conditions are shown in Figure 2. In all cases, we observe that whether weight decay was applied to the top (fully connected) layer did not have a significant impact; whether it was applied to the reamining (convolution) layers explained most of the generalization effect. This supports the effective learning rate hypothesis.

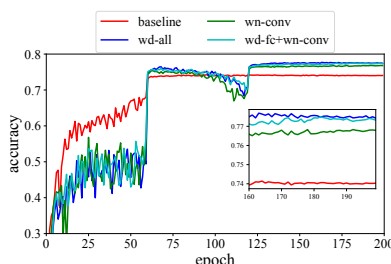

**Figure 4:** The curves of test accuracies of ResNet32 on CIFAR-100. To be noted, we use **wd** and **wn** to denote weight decay and weight normalization respectively.

We further tested this hypothesis using a simple experimental manipulation. Specifically, we trained a BN network without weight decay, but after each epoch, rescaled the weights in each layer to match that layer's norm from the corresponding epoch for the network with weight decay. This rescaling does not affect the network's predictions, and is equivalent to setting the effective learning rate to match the second network. As shown in Figure 4, this effective learning rate transfer scheme (wn-conv) eliminates almost the entire generalization gap; it is fully closed by also adding weight decay to the top layer (wd-fc+wn-conv). Hence, we conclude that for BN networks trained with SGD or Adam, weight decay achieves its regularization effect primarily through the effective learning rate.

### 4.2 MECHANISM II: APPROXIMATE JACOBIAN REGULARIZATION

In Section 3, we observed that when BN is disabled, weight decay has the strongest regularization effect when K-FAC is used as the optimizer. Hence, in this section we analyze the effect of weight decay for K-FAC with networks without BN. First, we show that in a certain idealized setting, K-FAC with weight decay regularizes the input-output Jacobian of the network. We then empirically investigate whether it behaves similarly for practical networks.

As discussed in Section 3, when the gradient updates are preconditioned by a matrix $\mathbf{C}$, weight decay can be viewed as approximate preconditioned gradient descent on the norm $\|\boldsymbol{\theta}\|_{\mathbf{C}}^2 = \boldsymbol{\theta}^\top \mathbf{C} \boldsymbol{\theta}$. This interpretation is only approximate because the exact gradient update requires differentiating through $\mathbf{C}$.[2] When $\mathbf{C}$ is taken to be the (exact) Gauss-Newton (GN) matrix $\mathbf{G}$, we obtain the Gauss-Newton norm $\|\boldsymbol{\theta}\|_{\mathbf{G}}^2 = \boldsymbol{\theta}^\top \mathbf{G}(\boldsymbol{\theta}) \boldsymbol{\theta}$. Similarly, when $\mathbf{C}$ is taken to be the K-FAC approximation to $\mathbf{G}$, we obtain what we term the *K-FAC Gauss-Newton norm*.

These norms are interesting from a regularization perspective. First, under certain conditions, they are proportional to the average $L_2$ norm of the network's outputs. Hence, the regularizer ought to make the network's predictions less extreme. This is summarized by the following results:

**Lemma 1** (Gradient structure)**.** *For a feed-forward neural network of depth $L$ with ReLU activation function and no biases, the network's outputs are related to the input-output Jacobian and parameter-output Jacobian as follows:*

$$
\begin{aligned}
f_{\boldsymbol{\theta}}(\mathbf{x}) &= \nabla_{\mathbf{x}} f_{\boldsymbol{\theta}}(\mathbf{x})^\top \mathbf{x} = \mathbf{J}_{\mathbf{x}} \mathbf{x} \\
&= \frac{1}{L+1} \nabla_{\boldsymbol{\theta}} f_{\boldsymbol{\theta}}(\mathbf{x})^\top \boldsymbol{\theta} = \frac{1}{L+1} \mathbf{J}_{\boldsymbol{\theta}} \boldsymbol{\theta}.
\end{aligned}
\tag{10}
$$

**Lemma 2** (Gauss-Newton Norm)**.** *Under the same assumptions of Lemma 1, we observe:*

$$
\|\boldsymbol{\theta}\|_{\mathbf{G}}^2 = (L+1)^2 \mathbb{E}\left[\|f_{\boldsymbol{\theta}}(\mathbf{x})\|^2\right].
\tag{11}
$$

*If we further restrict the network to be a deep linear neural network, we have K-FAC Gauss-Newton norm as follows:*

$$
\|\boldsymbol{\theta}\|_{\mathbf{G}_{\mathrm{K-FAC}}}^2 = (L+1) \mathbb{E}\left[\|f_{\boldsymbol{\theta}}(\mathbf{x})\|^2\right].
\tag{12}
$$

---

[2]We show in Appendix E that this interpretation holds exactly in the case of Gauss-Newton norm.

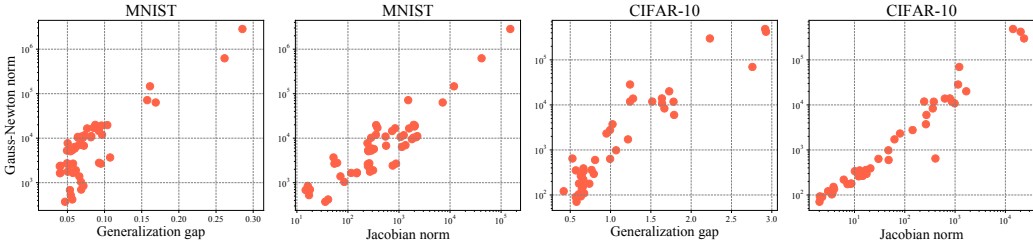

**Figure 5:** Relationship between K-FAC GN norm and Jacobian norm for practical deep neural networks. Each point corresponds to a network trained to 100% training accuracy. Even for (nonlinear) classification networks, the K-FAC GN norm is highly correlated with both the squared Frobenius norm of the input-output Jacobian and the generalization gap.

Using these results, we show that for linear networks[3] with whitened inputs, the (K-FAC) Gauss-Newton norm is proportional to the squared Frobenius norm of the input-output Jacobian. This is interesting from a regularization perspective, since Novak et al. (2018) found the norm of the input-output Jacobian to be consistently coupled to generalization performance.

**Theorem 1** (Approximate Jacobian norm). *For a deep linear network of depth L without biases, if we further assume that $\mathbb{E}[\mathbf{x}] = \mathbf{0}$ and $\mathrm{Cov}(\mathbf{x}) = \mathbf{I}$, then:*

$$\|\boldsymbol{\theta}\|_{\mathbf{G}}^2 = (L+1)^2 \|\mathbf{J_x}\|_{\mathrm{Frob}}^2 \tag{13}$$

*and*

$$\|\boldsymbol{\theta}\|_{\mathbf{G}_{\mathrm{K-FAC}}}^2 = (L+1) \|\mathbf{J_x}\|_{\mathrm{Frob}}^2. \tag{14}$$

*Proof.* It follows from Lemma 2 that $\|\boldsymbol{\theta}\|_{\mathbf{G}}^2 = (L+1)^2 \, \mathbb{E}\left[\|f_{\boldsymbol{\theta}}(\mathbf{x})\|^2\right]$. By Lemma 1, we have

$$\mathbb{E}\left[\|f_{\boldsymbol{\theta}}(\mathbf{x})\|^2\right] = \mathbb{E}\left[\mathbf{x}^\top \mathbf{J_x}^\top \mathbf{J_x} \mathbf{x}\right] = \mathbb{E}\left[\mathrm{tr}\,\mathbf{J_x}^\top \mathbf{J_x} \mathbf{x}\mathbf{x}^\top\right].$$

When the network is linear, the input-output Jacobian $\mathbf{J_x}$ is independent of the input $\mathbf{x}$. Then we use the assumption of whitened inputs:

$$\|\boldsymbol{\theta}\|_{\mathbf{G}}^2 = (L+1)^2 \, \mathbb{E}\left[\mathrm{tr}\,\mathbf{J_x}^\top \mathbf{J_x} \mathbf{x}\mathbf{x}^\top\right] = (L+1)^2 \, \mathrm{tr}\,\mathbf{J_x}^\top \mathbf{J_x} \mathbb{E}[\mathbf{x}\mathbf{x}^\top] = (L+1)^2 \|\mathbf{J_x}\|_{\mathrm{Frob}}^2.$$

The proof for K-FAC Gauss-Newton norm follows immediately with equation 12. □

While the equivalence between the (K-FAC) GN norm and the Jacobian norm holds only for linear networks, we note that linear networks have been useful for understanding the dynamics of neural net training more broadly (e.g. Saxe et al. (2013)). Hence, Jacobian regularization may help inform our understanding of weight decay in practical (nonlinear) networks.

To test whether the K-FAC GN norm correlates with the Jacobian norm for practical networks, we trained feed-forward networks with a variety optimizers on both MNIST (LeCun et al., 1998) and CIFAR-10. For MNIST, we used simple fully-connected networks with different depth and width. For CIFAR-10, we adopted the VGG family (From VGG11 to VGG19). We defined the generalization gap to be the difference between training and test loss. Figure 5 shows the relationship of the Jacobian norm to the K-FAC GN norm and to generalization gap for these networks. We observe that the Jacobian norm correlates strongly with the generalization gap (consistent with Novak et al. (2018)) and also with the K-FAC GN norm. Hence, Theorem 1 can inform the regularization of nonlinear networks.

**Table 2:** Squared Frobenius norm of the input-output Jacobian matrix. K-FAC-G with weight decay significantly reduces the Jacobian norm.

| Optimizer | VGG16 | | ResNet32 | |
|---|---|---|---|---|
| | | WD | | WD |
| SGD | 564 | 142 | 2765 | 1074 |
| K-FAC-G | 498 | 51.44 | 2115 | 64.16 |

To test if K-FAC with weight decay reduces the Jacobian norm, we compared the Jacobian norms at the end of training for networks with and without weight decay. As shown in Table 2, weight decay reduced the Jacboian norm by a much larger factor when K-FAC was used as the optimizer than when SGD was used as the optimizer.

---

[3] For K-FAC Gauss-Newton norm, we need the network to be linear due to the fact that K-FAC approximation is exact only for deep linear networks (Bernacchia et al., 2018).

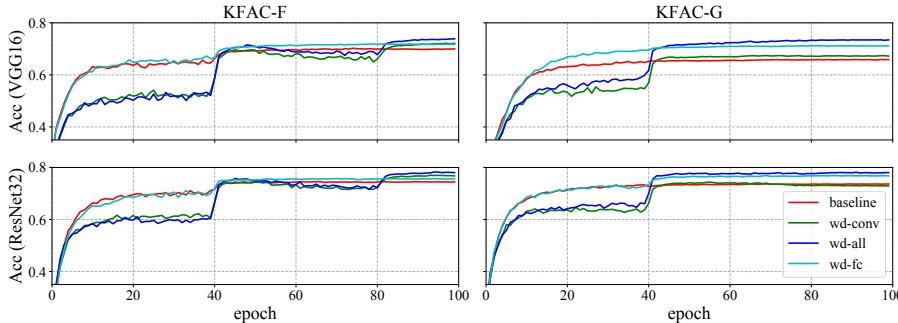

**Figure 6:** Test accuracy as a function of training epoch for K-FAC on CIFAR-100 with different weight decay regularization schemes. baseline is the model without weight decay regularization; wd-conv is the model with weight decay applied to all convolutional layers; wd-all is the model with weight decay applied to all layers; wd-fc is the model with weight decay applied to the last layer (fc). Consistent with the Jacobian regularization hypothesis, applying weight decay to the non-BN layers have the largest regularization effect. However, applying weight decay to the BN layers also lead to noticeable gains.

Our discussion so far as focused on the GN version of K-FAC. Recall that, in many cases, the Fisher information matrix differs from the GN matrix only in that it accounts for the output layer Hessian. Hence, this analysis may help inform the behavior of K-FAC-F as well. We also note that $\|\boldsymbol{\theta}\|_{\mathbf{F}}^2$, the Fisher-Rao norm, has been proposed as a complexity measure for neural networks (Liang et al., 2017). Hence, unlike in the case of SGD and Adam for BN networks, we interpret K-FAC with weight decay as constraining the capacity of the network.

### 4.3 MECHANISM III: SMALLER EFFECTIVE DAMPING PARAMETER

We now return our attention to the setting of architectures with BN. The Jacobian regularization mechanism from Section 4.2 does not apply in this case, since rescaling the weights results in an equivalent network, and therefore does not affect the input-output Jacobian. Similarly, if the network is trained with K-FAC, then the effective learning rate mechanism from Section 4.1 also does not apply because the K-FAC update is invariant to affine reparameterization (Luk & Grosse, 2018) and therefore not affected by the scaling of the weights. More precisely, for a layer with BN, the curvature matrix $\mathbf{C}$ (either the Fisher matrix or the GN matrix) has the following property:

$$\mathbf{C}(\boldsymbol{\theta}_l) = \frac{1}{\|\boldsymbol{\theta}_l\|_2^2}\mathbf{C}(\hat{\boldsymbol{\theta}}_l), \tag{15}$$

where $\hat{\boldsymbol{\theta}}_l = \boldsymbol{\theta}_l/\|\boldsymbol{\theta}_l\|_2$ as in Section 4.1. Hence, the $\|\boldsymbol{\theta}_l\|_2^2$ factor in the preconditioner counteracts the $\|\boldsymbol{\theta}_l\|_2^{-2}$ factor in the effective learning rate, resulting in an equivlaent effective learning rate regardless of the norm of the weights.

These observations raise the question of *whether it is still useful to apply weight decay to BN layers when using K-FAC*. To answer this question, we repeated the experiments in Figure 2 (applying weight decay to subsets of the layers), but with K-FAC as the optimizer. The results are summarized in Figure 6. Applying it to the non-BN layers had the largest effect, consistent with the Jacobian regularization hypothesis. However, applying weight decay to the BN layers also led to significant gains, especially for K-FAC-F.

The reason this does not contradict the K-FAC invariance property is that practical K-FAC implementations dampen the updates (like many second-order optimziers) by adding a multiple of the identity matrix to the curvature before inversion. According to equation 15, as the norm of the weights gets larger, $\mathbf{C}$ gets smaller, and hence the damping term comes to dominate the preconditioner. Mathematically, we can understand this effect by deriving the following update rule for the normalized weights $\hat{\boldsymbol{\theta}}$ (see Appendix D for proof):

$$\hat{\boldsymbol{\theta}}_l^{t+1} \leftarrow \hat{\boldsymbol{\theta}}_l^t - \eta(\mathbf{I} - \hat{\boldsymbol{\theta}}_l^t\hat{\boldsymbol{\theta}}_l^{t^\top})(\mathbf{C}(\hat{\boldsymbol{\theta}}_l^t) + \|\boldsymbol{\theta}_l^t\|_2^2\lambda\mathbf{I})^{-1}\nabla_{\boldsymbol{\theta}_l}\mathcal{L}(\hat{\boldsymbol{\theta}}^t) + O(\eta^2), \tag{16}$$

where $\lambda$ is the damping parameter. Hence, for large $\mathbf{C}(\hat{\boldsymbol{\theta}}_l)$ or small $\|\boldsymbol{\theta}_l\|$, the update is close to the idealized second-order update, while for small enough $\mathbf{C}(\hat{\boldsymbol{\theta}}_l)$ or large enough $\|\boldsymbol{\theta}_l\|$, K-FAC effectively becomes a first-order optimizer. Hence, by keeping the weights small, weight decay helps K-FAC to retain its second-order properties.

Most implementations of K-FAC keep the damping parameter $\lambda$ fixed throughout training. Therefore, it would be convenient if $\mathbf{C}(\hat{\boldsymbol{\theta}}_l)$ and $\|\boldsymbol{\theta}_l\|$ do not change too much during training, so that a single value of $\lambda$ can work well throughout training. Interestingly, the norm of the GN matrix appears to be much more stable than the norm of the Fisher matrix. Figure 7 shows the norms of the Fisher matrix $\mathbf{F}(\hat{\boldsymbol{\theta}}_l)$ and GN matrix $\mathbf{G}(\hat{\boldsymbol{\theta}}_l)$ of the normalized weights for the first layer of a CIFAR-10 network throughout training. While the norm of $\mathbf{F}(\hat{\boldsymbol{\theta}}_l)$ decays by 4 orders of magnitude over the first 50 epochs, the norm of $\mathbf{G}(\hat{\boldsymbol{\theta}}_l)$ increases by only a factor of 2.

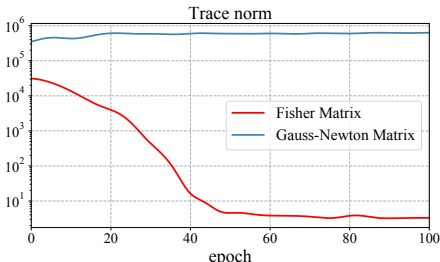

**Figure 7:** Trace norm of Fisher matrix and Gauss-Newton matrix of the first layer (Normalized) of ResNet32. The model was trained on CIFAR-10 with K-FAC-F and BN.

The explanation for this is as follows: in a classification task with cross-entropy loss, the Fisher matrix is equivalent to the generalized GN matrix $\mathbb{E}[\mathbf{J}_{\boldsymbol{\theta}}^\top \mathbf{H}_\ell \mathbf{J}_{\boldsymbol{\theta}}]$ (see Section 2). This differs from the GN matrix $\mathbb{E}[\mathbf{J}_{\boldsymbol{\theta}}^\top \mathbf{J}_{\boldsymbol{\theta}}]$ only in that it incudes the output layer Hessian $\mathbf{H}_\ell = \mathrm{diag}(\mathbf{p}) - \mathbf{p}\mathbf{p}^\top$, where $\mathbf{p}$ is the vector of estimated class probabilities. It is easy to see that $\mathbf{H}_\ell$ goes to zero as $\mathbf{p}$ collapses to one class, as is the case for tasks such as CIFAR-10 and CIFAR-100 where networks typically achieve perfect training accuracy. Hence, we would expect $\mathbf{F}$ to get much smaller over the course of training, consistent with Figure 7.

To summarize, when K-FAC is applied to BN networks, it can be advantageous to apply weight decay even to layers with BN, even though this appears unnecessary based on invariance considerations. The reason is that weight decay reduces the effective damping, helping K-FAC to retain its second-order properties. This effect is stronger for K-FAC-F than for K-FAC-G because the Fisher matrix shrinks dramatically over the course of training.

## 5 DISCUSSION

Despite its long history, weight decay regularization remains poorly understood. We've identified three distinct mechanisms by which weight decay improves generalization, depending on the architecture and optimization algorithm: increasing the effective learning rate, reducing the Jacobian norm, and reducing the effective damping parameter. We would not be surprised if there remain additional mechanisms we have not found.

The dynamics of neural net training is incredibly complex, and it can be tempting to simply do what works and not look into why. But we think it is important to at least sometimes dig deeper to determine exactly why an algorithm has the effect that it does. Some of our analysis may seem mundane, or even tedious, as the interactions between different hyperparameters are not commonly seen as a topic worthy of detailed scientific study. But our experiments highlight that the dynamics of the norms of weights and curvature matrices, and their interaction with optimization hyperparameters, can have a substantial impact on generalization. We believe these effects deserve more attention, and would not be surprised if they can help explain the apparent success or failure of other neural net design choices. We also believe our results highlight the need for automatic adaptation of optimization hyperparameters, to eliminate potential experimental confounds and to allow researchers and practitioners to focus on higher level design issues.

## 6 ACKNOWLEDGEMENT

We thank Jimmy Ba, David Duvenaud, Kevin Luk, Maxime Gazeau, and Behnam Neyshabur for helpful discussions, and Tianqi Chen and Shengyang Sun for their feedback on early drafts. GZ was funded by an MRIS Early Researcher Award.

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

## A    EXPERIMENTS DETAILS

Throughout the paper, we perform experiments on image classification with three different datasets, MNIST (LeCun et al., 1998), CIFAR-10 and CIFAR-100 (Krizhevsky & Hinton, 2009). For MNIST, we use simple fully-connected networks with different depth and width. For CIFAR-10 and CIFAR-100, we use VGG16 (Simonyan & Zisserman, 2014) and ResNet32 (He et al., 2016). To make the network more flexible, we widen all convolutional layers in ResNet32 by a factor of 4, according to Zagoruyko & Komodakis (2016).

We investigate three different optimization methods, including Stochastic Gradient Descent (SGD), Adam (Kingma & Ba, 2014) and K-FAC (Martens & Grosse, 2015). In K-FAC, two different curvature matrices are studied, including Fisher information matrix and Gauss-Newton matrix.

In default, batch size 128 is used unless stated otherwise. In SGD and Adam, we train the networks with a budge of 200 epochs and decay the learning rate by a factor of 10 every 60 epochs for batch sizes of 128 and 640, and every 80 epochs for the batch size of 2K. Whereas we train the networks only with 100 epochs and decay the learning rate every 40 epochs in K-FAC. Additionally, the curvature matrix is updated by running average with re-estimation every 10 iterations and the inverse operator is amortized to 100 iterations. For K-FAC, we use fixed damping term $1e^{-3}$ unless state otherwise. For each algorithm, best hyperparameters (learning rate and regularization factor) are selected using grid search on held-out 5k validation set. For the large batch setting, we adopt the same strategies in Hoffer et al. (2017) for adjusting the search range of hyperparameters. Finally, we retrain the model with both training data and validation data.

## B    GRADIENT STRUCTURE IN NEURAL NETWORKS (LEMMA 1)

*Claim.* For a feed-forward neural network of depth $L$ with ReLU activation function and no biases, one has the following property:

$$f_{\boldsymbol{\theta}}(\mathbf{x}) = \nabla_{\mathbf{x}} f_{\boldsymbol{\theta}}(\mathbf{x})^{\top} \mathbf{x} = \mathbf{J}_{\mathbf{x}} \mathbf{x}$$
$$= \frac{1}{L+1} \nabla_{\boldsymbol{\theta}} f_{\boldsymbol{\theta}}(\mathbf{x})^{\top} \boldsymbol{\theta} = \frac{1}{L+1} \mathbf{J}_{\boldsymbol{\theta}} \boldsymbol{\theta}$$

The key observation of Lemma 1 is that rectified neural networks are piecewise linear up to the output $f_{\boldsymbol{\theta}}(\mathbf{x})$. And ReLU activation function satisfies the property $\sigma(\mathbf{z}) = \sigma'(\mathbf{z})\mathbf{z}$.

*Proof.* For convenience, we introduce some notations here. Let $\mathbf{z}_{L+1}$ denotes output logits $f_{\boldsymbol{\theta}}(\mathbf{x})$, $\mathbf{z}_l$ the output $l$-th layer. Similarly, we define $\mathbf{a}_l = \sigma(\mathbf{z}_l)$ and $\mathbf{a}_0 = \mathbf{x}$. By definition, it is easy to see that

$$\mathbf{z}_{l+1} = \mathbf{W}_l \mathbf{a}_l = \frac{\partial \mathbf{z}_{l+1}}{\partial \mathbf{a}_l^{\top}} \mathbf{a}_l$$
$$= \frac{\partial \mathbf{z}_{l+1}}{\partial \mathbf{a}_l^{\top}} \frac{\partial \mathbf{a}_l}{\partial \mathbf{z}_l^{\top}} \mathbf{z}_l$$
$$= \frac{\partial \mathbf{z}_{l+1}}{\partial \mathbf{z}_l^{\top}} \mathbf{z}_l$$

By induction, we conclude that $f_{\boldsymbol{\theta}}(\mathbf{x}) = \nabla_{\mathbf{x}} f_{\boldsymbol{\theta}}(\mathbf{x})^{\top} \mathbf{x} = \mathbf{J}_{\mathbf{x}} \mathbf{x}$.

On the other side, we have

$$\mathbf{z}_{l+1} = \mathbf{W}_l \mathbf{a}_l = \sum_{i,j} \frac{\partial \mathbf{z}_{l+1}}{\partial \mathbf{W}_l^{i,j}} \mathbf{W}_l^{i,j}$$

According to equation B, $\mathbf{z}_{L+1} = \frac{\partial \mathbf{z}_{L+1}}{\partial \mathbf{z}_{L+1}} \mathbf{z}_{L+1}^{\top}$, therefore we get

$$\mathbf{z}_{L+1} = \sum_{i,j} \frac{\partial \mathbf{z}_{L+1}}{\partial \mathbf{W}_l^{i,j}} \mathbf{W}_l^{i,j}$$

Summing over all the layers, we conclude the following equation eventually:

$$(L+1)f_{\boldsymbol{\theta}}(\mathbf{x}) = \sum_l \sum_{i,j} \frac{\partial \mathbf{z}_{L+1}}{\partial \mathbf{W}_l^{i,j}} \mathbf{W}_l^{i,j} = \nabla_{\boldsymbol{\theta}} f_{\boldsymbol{\theta}}(\mathbf{x})^\top \boldsymbol{\theta} = \mathbf{J}_{\boldsymbol{\theta}} \boldsymbol{\theta}$$

$\square$

## C  PROOF OF LEMMA 2

*Claim.* For a feed-forward neural network of depth $L$ with ReLU activation function and no biases, we observe:

$$\|\boldsymbol{\theta}\|_{\mathbf{G}}^2 = (L+1)^2 \mathbb{E}\left[\|f_{\boldsymbol{\theta}}(\mathbf{x})\|^2\right]$$

Furthermore, if we restrict the network to be linear with only fully-connected layers, we have K-FAC Gauss-Newton norm as follows

$$\|\boldsymbol{\theta}\|_{\mathbf{G}_{\mathrm{K-FAC}}}^2 = (L+1)\mathbb{E}\left[\|f_{\boldsymbol{\theta}}(\mathbf{x})\|^2\right]$$

*Proof.* We first prove the equality $\|\boldsymbol{\theta}\|_{\mathbf{G}}^2 = (L+1)^2 \mathbb{E}\left[\|f_{\boldsymbol{\theta}}(\mathbf{x})\|^2\right]$. Using the definition of the Gauss-Newton norm in equation 3, we have

$$\|\boldsymbol{\theta}\|_{\mathbf{G}}^2 = \mathbb{E}\left[\boldsymbol{\theta}^\top \mathbf{J}_{\boldsymbol{\theta}}^\top \mathbf{J}_{\boldsymbol{\theta}} \boldsymbol{\theta}\right] = \mathbb{E}\left[\|\mathbf{J}_{\boldsymbol{\theta}} \boldsymbol{\theta}\|^2\right]$$

From Lemma 1, we have

$$\mathbf{J}_{\boldsymbol{\theta}} \boldsymbol{\theta} = (L+1)f_{\boldsymbol{\theta}}(\mathbf{x}) = (L+1)\mathbf{J}_{\mathbf{x}} \mathbf{x}$$

Combining above equalities, we arrive at the conclusion.

For second part $\|\boldsymbol{\theta}\|_{\mathbf{G}_{\mathrm{K-FAC}}}^2 = (L+1)\mathbb{E}\left[\|f_{\boldsymbol{\theta}}(\mathbf{x})\|^2\right]$, we note that kronecker-product is exact under the condition that the network is linear (Bernacchia et al., 2018), which means $\mathbf{G}_{\mathrm{K-FAC}}$ is the diagonal block version of Gauss-Newton matrix $\mathbf{G}$. Therefore, we have

$$\|\boldsymbol{\theta}\|_{\mathbf{G}_{\mathrm{K-FAC}}}^2 = \sum_l \mathbb{E}\left[\boldsymbol{\theta}_l^\top \mathbf{J}_{\boldsymbol{\theta}_l}^\top \mathbf{J}_{\boldsymbol{\theta}_l} \boldsymbol{\theta}_l\right]$$

According to Lemma 1, we have $\mathbb{E}\left[\boldsymbol{\theta}_l^\top \mathbf{J}_{\boldsymbol{\theta}_l}^\top \mathbf{J}_{\boldsymbol{\theta}_l} \boldsymbol{\theta}_l\right] = \mathbb{E}\left[\|f_{\boldsymbol{\theta}}(\mathbf{x})\|^2\right]$, therefore we conclude that

$$\|\boldsymbol{\theta}\|_{\mathbf{G}_{\mathrm{K-FAC}}}^2 = (L+1)\mathbb{E}\left[\|f_{\boldsymbol{\theta}}(\mathbf{x})\|^2\right]$$

$\square$

## D  DERIVATION OF EQUATION 16

*Claim.* During training, the weight direction $\hat{\boldsymbol{\theta}}_l^t = \boldsymbol{\theta}_l^t / \|\boldsymbol{\theta}_l^t\|_2$ is updated according to

$$\hat{\boldsymbol{\theta}}_{t+1} \leftarrow \hat{\boldsymbol{\theta}}_t - \eta(\mathbf{I} - \hat{\boldsymbol{\theta}}_t \hat{\boldsymbol{\theta}}_t^\top)(\mathbf{C}(\hat{\boldsymbol{\theta}}_t) + \|\boldsymbol{\theta}_t\|_2^2 \lambda \mathbf{I})^{-1} \nabla \mathcal{L}(\hat{\boldsymbol{\theta}}_t) + O(\eta^2)$$

*Proof.* Natural gradient update is given by

$$\boldsymbol{\theta}_{t+1} \leftarrow \boldsymbol{\theta}_t - \eta(\mathbf{C}(\boldsymbol{\theta}_t) + \lambda \mathbf{I})^{-1} \nabla \mathcal{L}(\boldsymbol{\theta}_t)$$

Denote $\rho_t = \|\boldsymbol{\theta}_t\|_2$. Then we have

$$\rho_{t+1}^2 = \rho_t^2 - 2\eta \rho_t^2 \hat{\boldsymbol{\theta}}_t^\top (\mathbf{C}(\hat{\boldsymbol{\theta}}_t) + \lambda \rho_t^2 \mathbf{I})^{-1} \nabla \mathcal{L}(\hat{\boldsymbol{\theta}}_t) + \eta^2 \rho_t^2 \|(\mathbf{C}(\hat{\boldsymbol{\theta}}_t) + \lambda \rho_t^2 \mathbf{I})^{-1} \nabla \mathcal{L}(\hat{\boldsymbol{\theta}}_t)\|_2^2$$

and therefore

$$\rho_{t+1} = \rho_t \sqrt{1 - 2\eta \hat{\boldsymbol{\theta}}_t^\top (\mathbf{C}(\hat{\boldsymbol{\theta}}_t) + \lambda \rho_t^2 \mathbf{I})^{-1} \nabla \mathcal{L}(\hat{\boldsymbol{\theta}}_t) + \eta^2 \|(\mathbf{C}(\hat{\boldsymbol{\theta}}_t) + \lambda \rho_t^2 \mathbf{I})^{-1} \nabla \mathcal{L}(\hat{\boldsymbol{\theta}}_t)\|_2^2}$$

$$= \rho_t(1 - \eta \hat{\boldsymbol{\theta}}_t^\top (\mathbf{C}(\hat{\boldsymbol{\theta}}_t) + \lambda \rho_t^2 \mathbf{I})^{-1} \nabla \mathcal{L}(\hat{\boldsymbol{\theta}}_t)) + O(\eta^2)$$

Additionally, we can rewrite the natural gradient update as follows

$$\rho_{t+1} \hat{\boldsymbol{\theta}}_{t+1} = \rho_t \hat{\boldsymbol{\theta}}_t - \eta \rho_t (\mathbf{C}(\hat{\boldsymbol{\theta}}_t) + \lambda \rho_t^2 \mathbf{I})^{-1} \nabla \mathcal{L}(\hat{\boldsymbol{\theta}}_t)$$

And therefore,

$$
\begin{aligned}
\hat{\boldsymbol{\theta}}_{t+1} &= \frac{\rho_t}{\rho_{t+1}} \left( \hat{\boldsymbol{\theta}}_t - \eta (\mathbf{C}(\hat{\boldsymbol{\theta}}_t) + \lambda \rho_t^2 \mathbf{I})^{-1} \nabla \mathcal{L}(\hat{\boldsymbol{\theta}}_t) \right) \\
&= \left( 1 + \eta \hat{\boldsymbol{\theta}}_t^\top (\mathbf{C}(\hat{\boldsymbol{\theta}}_t) + \lambda \rho_t^2 \mathbf{I})^{-1} \nabla \mathcal{L}(\hat{\boldsymbol{\theta}}_t) \right) \left( \hat{\boldsymbol{\theta}}_t - \eta (\mathbf{C}(\hat{\boldsymbol{\theta}}_t) + \lambda \rho_t^2 \mathbf{I})^{-1} \nabla \mathcal{L}(\hat{\boldsymbol{\theta}}_t) \right) + O(\eta^2) \\
&= \hat{\boldsymbol{\theta}}_t - \eta (\mathbf{I} - \hat{\boldsymbol{\theta}}_t \hat{\boldsymbol{\theta}}_t^\top)(\mathbf{C}(\hat{\boldsymbol{\theta}}_t) + \|\boldsymbol{\theta}_t\|_2^2 \lambda \mathbf{I})^{-1} \nabla \mathcal{L}(\hat{\boldsymbol{\theta}}_t) + O(\eta^2)
\end{aligned}
$$

$\square$

## E  THE GRADIENT OF GAUSS-NEWTON NORM

For Gauss-Newton norm $\|\boldsymbol{\theta}\|_{\mathbf{G}}^2 = (L+1)^2 \mathbb{E}_{\mathbf{x}} \left[ \langle f_{\boldsymbol{\theta}}(\mathbf{x}), f_{\boldsymbol{\theta}}(\mathbf{x}) \rangle \right]$, its gradient has the following form:

$$
\frac{\partial \|\boldsymbol{\theta}\|_{\mathbf{G}}^2}{\partial \boldsymbol{\theta}} = 2(L+1)^2 \mathbb{E} \left[ \mathbf{J}_{\boldsymbol{\theta}}^\top f_{\boldsymbol{\theta}}(\mathbf{x}) \right] \tag{17}
$$

According to Lemma 1, we have $f_{\boldsymbol{\theta}}(\mathbf{x}) = \frac{1}{L+1} \mathbf{J}_{\boldsymbol{\theta}} \boldsymbol{\theta}$, therefore we can rewrite equation 17

$$
\begin{aligned}
\frac{\partial \|\boldsymbol{\theta}\|_{\mathbf{G}}^2}{\partial \boldsymbol{\theta}} &= 2(L+1) \mathbb{E} \left[ \mathbf{J}_{\boldsymbol{\theta}}^\top \mathbf{J}_{\boldsymbol{\theta}} \boldsymbol{\theta} \right] \\
&= 2(L+1) \mathbf{G} \boldsymbol{\theta}
\end{aligned} \tag{18}
$$

Surprisingly, the resulting gradient has the same form as the case where we take Gauss-Newton matrix as a constant of $\boldsymbol{\theta}$ up to a constant $(L+1)$.

## F  KRONECKER-FACTORED APPROXIMATE CURVATURE (K-FAC)

Martens & Grosse (2015) proposed K-FAC for performing efficient natural gradient optimization in deep neural networks. Following on that work, K-FAC has been adopted in many tasks (Wu et al., 2017; Zhang et al., 2017) to gain optimization benefits, and was shown to be amendable to distributed computation (Ba et al., 2016).

### F.1  BASIC IDEA OF K-FAC

As shown by Luk & Grosse (2018), K-FAC can be applied to general pullback metric, including Fisher metric and the Gauss-Newton metric. For convenience, we introduce K-FAC here using the Fisher metric.

Considering $l$-th layer in the neural network whose input activations are $\mathbf{a}_l \in \mathbb{R}^{n_1}$, weight $\mathbf{W}_l \in \mathbb{R}^{n_1 \times n_2}$, and output $\mathbf{s}_l \in \mathbb{R}^{n_2}$, we have $\mathbf{s}_l = \mathbf{W}_l^\top \mathbf{a}_l$. Therefore, weight gradient is $\nabla_{\mathbf{W}_l} \mathcal{L} = \mathbf{a}_l (\nabla_{\mathbf{s}_l} \mathcal{L})^\top$. With this gradient formula, K-FAC decouples this layer's fisher matrix $\mathbf{F}_l$ using mild approximations,

$$
\begin{aligned}
\mathbf{F}_l &= \mathbb{E} \left[ \operatorname{vec}\{\nabla_{\mathbf{W}_l} \mathcal{L}\} \operatorname{vec}\{\nabla_{\mathbf{W}_l} \mathcal{L}\}^\top \right] = \mathbb{E} \left[ \{\nabla_{\mathbf{s}_l} \mathcal{L}\}\{\nabla_{\mathbf{s}_l} \mathcal{L}\}^\top \otimes \mathbf{a}_l \mathbf{a}_l^\top \right] \\
&\approx \mathbb{E} \left[ \{\nabla_{\mathbf{s}_l} \mathcal{L}\}\{\nabla_{\mathbf{s}_l} \mathcal{L}\}^\top \right] \otimes \mathbb{E} \left[ \mathbf{a}_l \mathbf{a}_l^\top \right] = \mathbf{S}_l \otimes \mathbf{A}_l
\end{aligned} \tag{19}
$$

Where $\mathbf{A}_l = \mathbb{E} \left[ \mathbf{a} \mathbf{a}^\top \right]$ and $\mathbf{S}_l = \mathbb{E} \left[ \{\nabla_{\mathbf{s}} \mathcal{L}\}\{\nabla_{\mathbf{s}} \mathcal{L}\}^\top \right]$. The approximation above assumes independence between $\mathbf{a}$ and $\mathbf{s}$, which proves to be accurate in practice. Further, assuming between-layer independence, the whole fisher matrix $\mathbf{F}$ can be approximated as block diagonal consisting of layer-wise fisher matrices $\mathbf{F}_l$. Decoupling $\mathbf{F}_l$ into $\mathbf{A}_l$ and $\mathbf{S}_l$ not only avoids the memory issue saving $\mathbf{F}_l$, but also provides efficient natural gradient computation.

$$
\mathbf{F}_l^{-1} \operatorname{vec}\{\nabla_{\mathbf{W}_l} \mathcal{L}\} = \mathbf{S}_l^{-1} \otimes \mathbf{A}_l^{-1} \operatorname{vec}\{\nabla_{\mathbf{W}_l} \mathcal{L}\} = \operatorname{vec}[\mathbf{A}_l^{-1} \nabla_{\mathbf{W}_l} \mathcal{L} \mathbf{S}_l^{-1}] \tag{20}
$$

As shown by equation 20, computing natural gradient using K-FAC only consists of matrix transformations comparable to size of $\mathbf{W}_l$, making it very efficient.

### F.2  PSEUDO CODE OF K-FAC

---

**Algorithm 1** K-FAC with $L_2$ regularization and K-FAC with weight decay. Subscript $l$ denotes layers, $\mathbf{w}_l = \mathrm{vec}(\mathbf{W}_l)$. We assume zero momentum for simplicity.

---

**Require:** $\eta$: stepsize
**Require:** $\beta$: weight decay
**Require:** stats and inverse update intervals $T_{\mathrm{stats}}$ and $T_{\mathrm{inv}}$
   $k \leftarrow 0$ and initialize $\{\mathbf{W}_l\}_{l=1}^{L}, \{\mathbf{S}_l\}_{l=1}^{L}, \{\mathbf{A}_l\}_{l=1}^{L}$
   **while** stopping criterion not met **do**
     $k \leftarrow k + 1$
     **if** $k \equiv 0 \pmod{T_{\mathrm{stats}}}$ **then**
       Update the factors $\{\mathbf{S}_l\}_{l=1}^{L}, \{\mathbf{A}_l\}_{l=0}^{L-1}$ with moving average
     **end if**
     **if** $k \equiv 0 \pmod{T_{\mathrm{inv}}}$ **then**
       Calculate the inverses $\{[\mathbf{S}_l]^{-1}\}_{l=1}^{L}, \{[\mathbf{A}_l]^{-1}\}_{l=0}^{L-1}$
     **end if**
     $\mathbf{V}_l = \nabla_{\mathbf{W}_l} \log p(y|\mathbf{x}, \mathbf{w}) + \beta \cdot \mathbf{W}_l$
     $\mathbf{W}_l \leftarrow \mathbf{W}_l - \left( \eta [\mathbf{A}_l]^{-1} \mathbf{V}_l [\mathbf{S}_l]^{-1} + \beta \cdot \mathbf{W}_l \right)$
   **end while**

---

# G   ADDITIONAL RESULTS

## G.1   LARGE-BATCH TRAINING

It has been shown that K-FAC scales very favorably to larger mini-batches compared to SGD, enjoying a nearly linear relationship between mini-batch size and per-iteration progress for medium-to-large sized mini-batches (Martens & Grosse, 2015; Ba et al., 2016). However, Keskar et al. (2016) showed that large-batch methods converge to sharp minima and generalize worse. In this subsection, we measure the generalization performance of K-FAC with large batch training and analyze the effect of weight decay.

In Table 3, we compare K-FAC with SGD using different batch sizes. In particular, we interpolate between small-batch (BS128) and large-batch (BS2000). We can see that in accordance with previous works (Keskar et al., 2016; Hoffer et al., 2017) the move from a small-batch to a large-batch indeed incurs a substantial generalization gap. However, adding weight decay regularization to K-FAC almost close the gap on CIFAR-10 and cause much of the gap diminish on CIFAR-100. Surprisingly, the generalization gap of SGD also disappears with well-tuned weight decay regularization. Moreover, we observe that the training loss cannot decrease to zero if weight decay is not used, indicating weight decay may also speed up the training.

**Table 3:** Classification results with different batch sizes. **WD** denotes weight decay regularization. We tune weight decay factor and learning rate using held-out validation set.

| Dataset | Network | Method | BS128 | | BS640 | | BS2000 | |
|---|---|---|---|---|---|---|---|---|
| | | | | WD | | WD | | WD |
| CIFAR10 | VGG16 | SGD | 91.71 | 93.39 | 90.46 | 93.09 | 88.50 | 92.24 |
| | | K-FAC-F | 93.12 | 93.90 | 92.93 | 93.55 | 92.17 | 93.31 |
| | | K-FAC-G | 93.19 | 93.80 | 92.98 | 93.74 | 90.78 | 93.46 |
| CIFAR10 | ResNet32 | SGD | 92.95 | 95.14 | 91.68 | 94.45 | 89.70 | 94.68 |
| | | K-FAC-F | 93.80 | 95.35 | 92.30 | 94.79 | 91.15 | 94.43 |
| | | K-FAC-G | 93.44 | 95.04 | 91.80 | 94.73 | 90.02 | 94.85 |
| CIFAR100 | ResNet32 | SGD | 73.61 | 77.73 | 71.74 | 76.67 | 65.38 | 76.87 |
| | | K-FAC-F | 74.49 | 78.01 | 73.54 | 77.34 | 71.64 | 77.13 |
| | | K-FAC-G | 73.70 | 78.02 | 71.13 | 77.40 | 65.41 | 76.93 |

## G.2 THE CURVES OF TEST ACCURACIES

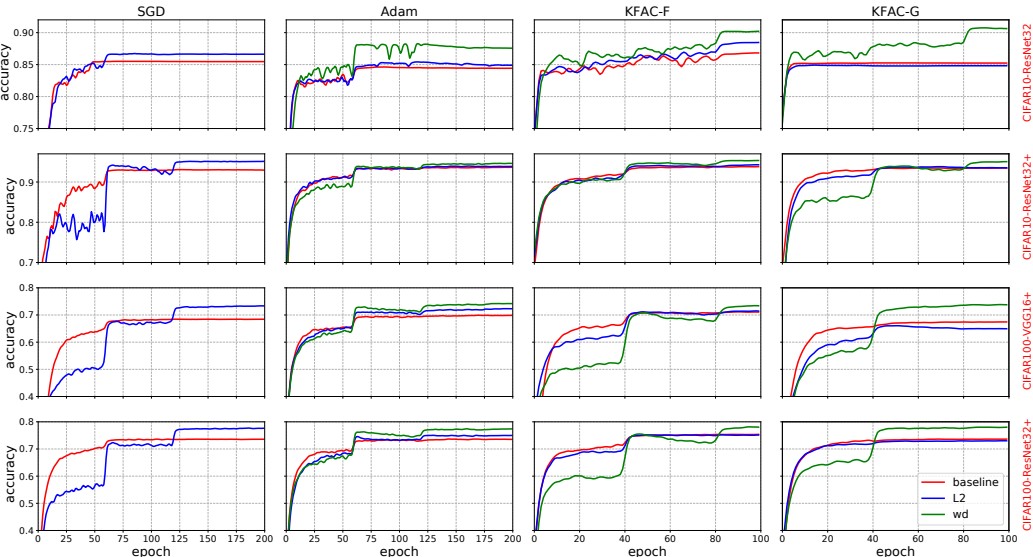

**Figure 8:** Test accuracy as a function of training epoch. We plot baseline vs $L_2$ regularization vs weight decay regularization on CIFAR-10 and CIFAR-100 datasets. The '+' denotes with BN and data augmentation. Note that training accuracies of all the models are $100\%$ in the end of the training. We smooth all the curves for visual clarity.

## G.3 OPTIMIZATION PERFORMANCE OF DIFFERENT OPTIMIZERS

While this paper mostly focus on generalization, we also report the convergence speed of different optimizers in deep neural networks; we report both per-epoch performance and wall-clock time performance.

We consider the task of image classification on CIFAR-10 (Krizhevsky & Hinton, 2009) dataset. The models we use consist of VGG16 (Simonyan & Zisserman, 2014) and ResNet32 (He et al., 2016). We compare our K-FAC-G, K-FAC-F with SGD, Adam (Kingma & Ba, 2014). We experiment with constant learning for K-FAC-G and K-FAC-F. For SGD and Adam, we set batch size as 128. For K-FAC, we use batch size of 640, as suggested by Martens & Grosse (2015).

In Figure 9, we report the training curves of different algorithms. Figure 9a show that K-FAC-G yields better optimization than other baselines in training loss per epoch. We highlight that the training loss decreases to 1e-4 within 10 epochs with K-FAC-G. Although K-FAC based algorithms take more time for each epoch, Figure 9b still shows wall-clock time improvements over the baselines.

In Figure 9c and 9d, we report similar results on the ResNet32. Note that we make the network wider with a widening factor of 4 according to Zagoruyko & Komodakis (2016). K-FAC-G outperforms both K-FAC-F and other baselines in term of optimization per epoch, and compute time.

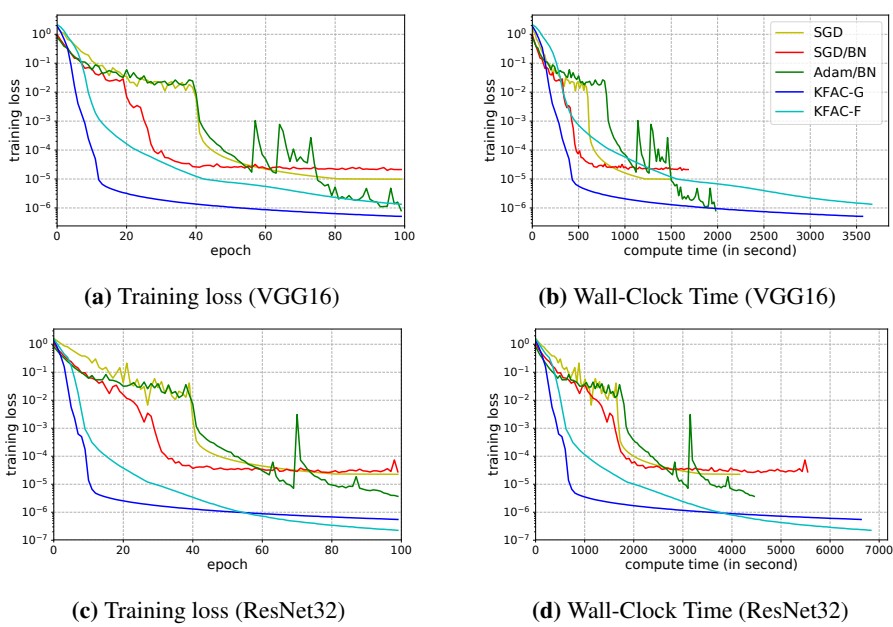

**(a)** Training loss (VGG16)

**(b)** Wall-Clock Time (VGG16)

**(c)** Training loss (ResNet32)

**(d)** Wall-Clock Time (ResNet32)

**Figure 9:** CIFAR-10 image classification task.

