# OpenReview forum: "Three Mechanisms of Weight Decay Regularization"
_ICLR.cc/2019/Conference_

### Official Review · AnonReviewer1 · 2018-10-31
**Nice insights about second order methods**

**Rating:** 7
**Confidence:** 5

**Review:**

This paper discusses the effect of weight decay on the training of deep network models with and without batch normalization and when using first/second order optimization methods.

First, it is discussed how weight decay affects the learning dynamics in networks with batch normalization when trained with SGD. The dominant generalization benefit due to weight decay comes from increasing the effective learning rate of parameters on which batch normalization is applied. The authors therefore hypothesize that a larger learning rate has a regularization effect.

Second, the role of weight decay is discussed when training with second order methods without batch normalization. Under the approximation of not differentiating the curvature matrix used in second order method, it is shown that using weight decay is equivalent to adding to the loss an L2 regularization in the metric space of the curvature matrix considered. It is then shown that if the curvature matrix is the Gauss-Newton matrix, this L2 regularization (and hence the weight decay) is equivalent to the Frobenius norm of the input-output Jacobian when the input has a spherical Gaussian distribution. Similar arguments are made about KFAC with Gauss-Newton norm. The generalization benefit due to weight decay in this case is claimed based on the recent paper by Novak et al 2018 which empirically shows a strong correlation between input-output Jacobian norm and generalization error.


Finally, the role of weight decay is discussed for second order methods when using batch normalization. In this case it is discussed for Gauss-Newton KFAC that the benefit mostly comes from the application of weight decay on the softmax layer and the effect of weight decay on other weights cancel out due to batch normalization. A comparison between Gauss-Newton KFAC and Fischer KFAC is also made. Thus the generalization benefit is presumably attributed to the second order properties of KFAC and a smaller norm of softmax layer weights.

Comments:
The paper is technically correct and proofs look good.

I have mixed comments about this paper. I find the analysis in section 4.2 and 4.3 which discuss about the role of weight decay for second order methods (with and without batch-norm) to be novel and insightful (described above).

But on the other hand, I feel section 4.1 is more of a discussion on existing work rather than novel contribution. Most of what is said, both analytically and experimentally, is a repetition of van Laarhoven 2017, except for a few details. It would have been interesting to carefully study the effect of weight decay on the gamma parameter of batch-norm which controls the complexity of the network along with the softmax layer weights as it was left for future work in van Laarhoven 2017. But instead the authors brush it under the carpet by saying they did not find the gamma and beta parameters to have significant impact on performance, and fixed them during training.  I also find the claim of section 4.1 to be a bit mis-leading because it is claimed that weight decay applied with SGD and batch normalization only has benefits due to batch-norm dynamics, and not due to complexity control even though in Fig 2 and 4, there is a noticeable difference between training without weight decay, and training with weight decay only on last layer. Furthermore, when hypothesizing the regularization effect of large learning rate in section 4.1, a large body of literature that has studied this effect has not been cited. Examples are [1], [2], [3].

I have other concerns which mainly stem from lack of clarity in writing:

1. In the line right above remark 1, it is not clear what “assumption” refer to. I am guessing the distribution of the input being spherical Gaussian?
2. In remark 1, regarding the claim about the equivalence of L2 norm of theta under Gauss-Newton metric and the Frobenius norm of input-output Jacobian, why does f_theta need to be a linear function without any non-linearity? I think the linearity part is only needed for the KFAC result.
3. In remark 1, what does it mean by “Furthermore, if G is approximated by KFAC”? For linear f_theta, given lemma 1 and theorem 1, the claimed equivalence always holds true, no?
4. In the 1st line of last paragraph of page 6, what are the general conditions under which the connection between Gauss-Newton norm and Jacobian norm does not hold true?
5. In figure 5, how are the different points in the plots achieved? By varying hyper-parameters?

A minor suggestion: in theorem 1 (and lemma 1), instead of assuming network has no bias, it can be said that the L2 regularization term does not have bias terms. This is more reasonable because bias terms have no effect on complexity and so it is reasonable to not apply weight decay on bias.

Overall I think the paper is good *if* section 4.1 is sorted out and writing (especially in section 4.2) is improved. For these reasons, I am currently giving a score of 6, but I will increase it if my concerns are addressed.

[1] a bayesian perspective on generalization and stochastic gradient descent
[2] Train longer, generalize better: closing the generalization gap in large batch training of neural networks
[3] Three Factors Influencing Minima in SGD

---

> ### Author Response · Authors · 2018-11-06
> **Response to AnonReviewer1**
>
> Thank you for the useful feedback. We have updated the paper (especially 4.2) taking into account several of your comments.
>
> Q1: Mechanism 1 is more of a discussion on existing work rather than novel contribution
> We agree that the argument of "effective learning rate" itself is not novel and has been observed by van Laarhoven 2017.
> However, we don't think the mechanism 1 is just a discussion of existing work. Particularly, van Laarhoven 2017 didn’t show any experiments that weight decay improves generalization performance. In Figure 2 of van Laarhoven 2017, they only showed that small learning rate is preferred when weight decay is applied. The important point we made is that weight decay actually improves the generalization performance even with well-tuned learning rate parameter and the gain of applying weight decay cannot be achieved by tunning the learning rate directly (we shouldn't ignore the interaction between the learning rate and weight decay).
>
> Furthermore, van Laarhoven 2017 was just talking about L2 regularization which is not equivalent to weight decay in adaptive gradient methods. We don't think the author realized the subtle difference between L2 regularization and weight decay. In the combination of L2 regularization and adaptive gradient methods, the argument of effective learning rate might not hold exactly since L2 regularization can affect both the scale and direction of the weights. In our paper, we extend the argument of "effective learning rate" to first-order optimization algorithms (including SGD and Adam) by identifying the subtle difference between L2 regularization and weight decay.
>
> Q2: The effect of weight decay on the gamma parameter of batch-norm.
> As discussed in van Laarhoven 2017, only the gamma of the last BN layer affects the complexity of the network. The role of it is quite similar to the scale of the last fully connected layer since you can always merge the gamma parameter into the last fc layer. In practice, the gain of regularizing the gamma parameter of the last BN layer is quite small which is consistent with our observation that regularizing the last fc layer gives marginal improvement. That's why we fixed the gamma parameter throughout the paper.
>
> Q3: In Figure 2 and 4, there is a noticeable difference between training without weight decay, and training with weight decay only on the last layer.
> In Figure 2, the gap is pretty small (<1%).
> In Figure 4, regularizing the last layer does help a little bit (~1%) while the improvement of regularizing conv layers is much larger (~3%).
> According to your suggestion, we revised our statements in 4.1 to make the arguments softer.
>
> Q4: In the line right above remark 1, what does “assumption” refer to?
> It does refer to spherical Gaussian input distribution. We have improved the writing for this part, it should be much clearer now.
>
> Q5: Regarding the equivalence of L2 norm of theta under Gauss-Newton metric and the Frobenius norm of input-output Jacobian, why does f_theta need to be a linear function without any non-linearity?
> That’s because we want the input-output Jacobian to be independent of the input x (which is not true for non-linear networks). Under this assumption, we can take J_x out of the expectation (see revised Theorem 1).
>
> Note: if the (all) input x has entries ±1 (so that xx^T is an identity matrix), then the assumption of f_theta being linear is not necessary. In that case, it is easy to show that the Gauss-Newton norm is proportional to the expectation of squared Jacobian norm over input distribution.
>
> Q6: In remark 1, what does it mean by “Furthermore, if G is approximated by KFAC”?
> This original claim is a little misleading, we have rewritten this part. Basically, when G is approximated by K-FAC (it's intractable to use exact G in practice), the K-FAC Gauss-Newton norm is still proportional to the squared Jacobian norm, but the constant becomes (L+1), not (L+1)**2.
>
> Q7: In the 1st line of the last paragraph of page 6, what are the general conditions under which the connection between Gauss-Newton norm and Jacobian norm does not hold true?
> If the network is not linear, then the connection will not hold exactly. We need the assumption of the network being linear so that the input-output Jacobian J_x is independent of the input x.
>
> Q8: In Figure 5, how are the different points in the plots achieved? By varying hyper-parameters?
> Sorry, we didn't explain Figure 5 clearly in the submitted version. Different points are achieved by varying optimizers and architectures (we mentioned that on page 7 of the updated version). Specifically, we trained feed-forward networks with a variety optimizers on both MNIST and CIFAR-10. For MNIST, we used simple fully-connected networks with different depth and width. For CIFAR-10, we adopted the VGG family (From VGG11 to VGG19).
>
> Q9: Missing citations
> Thank you for pointing out missing citations. We added multiple citations in the latest version.

---

> > ### Comment · AnonReviewer1 · 2018-11-13
> > **Comments**
> >
> > Q1: Agreed
> >
> > Q2: You are right about weight decay on gamma only affecting the complexity of the model due to the last layer which can be merged with the softmax layer weights (as also pointed out by van Laarhoven). May be mention this below Eq. 5 (while citing van Laarhoven) to remind the reader of this fact.
> >
> > Q3:
> > On page 6 (left of Figure 4), I recommend changing the sentence
> > "In all cases, we observe that whether weight decay was applied to the top (fully connected) layer did not appear to matter;"
> > to something like
> > "In all cases, we observe that whether weight decay was applied to the top (fully connected) layer did not have a significant impact;"
> >
> > Q4: OK
> >
> > Q5: Thank you for clarifying. I can see the technical mistake made in the 1st submission involving expectation over the input-output Jacobian for ReLU networks. However the current Theorem 1 on deep linear network makes the claim weak and the authors have used earlier work on deep linear networks as a justification.
> >
> > Q6,7,8,9: OK
> >
> > Comments:
> >
> > There were a few technical mistakes in the original submission that were overlooked by the reviewers and the authors have themselves identified and corrected them. However, these corrections have made the results for the second order methods weaker (section 4.2) since they apply to deep linear networks, which is a bit disappointing. But I still think this paper deserves to be read because 1. even though based on intuitions from deep linear networks, experiments are shown for deep non-linear networks confirming the insights drawn from them; 2. other sections have complementary analysis of weight decay for additional cases.
> >
> > (I have increased my original score by 1)

---

> > > ### Author Response · Authors · 2018-11-15
> > > **Re: Comments**
> > >
> > > Thank you for your new comments. We will update the paper according to your suggestions (Q2 and Q3).

---

### Official Review · AnonReviewer3 · 2018-11-01
**Solid work on understanding of weight decay regularization**

**Rating:** 7
**Confidence:** 4

**Review:**

This paper identifies and investigates three mechanisms of weight decay regularization. The authors consider weight decay for DNN architectures with/without BN and different types of optimization algorithms (SGD, Adam, and two versions of KFAC). The paper unravels insights on weight decay regularization effects, which cannot be explained only by traditional L2 regularization approach. This understanding is of high importance for the further development of regulations techniques for deep learning.

Strengths:
+ The authors draw connections between identified mechanisms and effects observed in prior work.
+ The authors provide both clear theoretical analysis and adequate experimental evidence supporting identified regularization mechanisms.
+ The paper is organized and written clearly.

I cannot point out any flaws in the paper. The only recommendation I would give is to discuss in more detail possible implications of the observed results for new methods of regularization in deep learning and potential directions for future work. It would emphasize the significance of the obtained results.

---

> ### Author Response · Authors · 2018-11-06
> **Response to AnonReviewer3**
>
> We thank the reviewer for the positive feedback.
>
> We have revised the conclusion section to discuss the observed results and potential new directions for future work.

---

> > ### Comment · AnonReviewer3 · 2018-12-04
> > **I stick to my rating**
> >
> > The authors have taken my comment into account in the new revision of the paper and adequately addressed issues pointed out by other reviewers. So, I keep my rating unchanged.

---

### Official Review · AnonReviewer2 · 2018-11-02
**Writing needs improvement; many handwavy explanations**

**Rating:** 6
**Confidence:** 4

**Review:**

I have read the author's response, and I would like to stick to my rating. From the authors' response on the convergence issue, the result from [1] does not directly apply since the activation function that the authors use in this paper is relu (not linear). Having said that, authors didn't find any issues empirically.

Q7: Yes, I agree that the result depends on the gradient structure of the relu activations. But my point was that, it is still a calculation that one has to carry out, and the insight we gain from the calculation seem computational: that one can regularize jacobian norm easily. True, but is that necessary? Or in other words, can we use techniques (not-so) recent  implicit regularization literature to analyze KFAC? I still think that the work is good, these are just my questions.
====

The paper investigates how weight decay (according to the authors, this is done by scaling weights at each iteration) can be used as a regularizer while using standard first order methods and KFAC. As far as I can see, the experimental conclusion seem pretty consistent with other papers that the authors themselves cite (for eg: Neelakantan et al. (2015);  Martens & Grosse, 2015.

In page 2, the authors mention the three different mechanisms by which weight decay has a regularizing effect. First, what is the definition of "effective learning rate"? If the authors mean that regularization just changes the learning rate in some case, that is true. In fact, it is only true while using l2-norm. I looked through the paper, and I couldn't find one. Similarly, I find point #1. to be confusing: why does reducing the scale of the weights increase the effective learning rate? (This confusion carries over to/remains in section 4.1.). The sentence starting (in point #1.) with "As evidence,", what is the evidence for? Is it for the previous statement that weight decay helps as a regularizer? Looking at Figure 1., Table 1., I can see that weight decay is actually helpful even with BN+D. In fact, the improvement provided by weight decay is uniform across the board.

The conclusion of mechanism 1 is that for layers with BN, weight decay is implicitly using higher learning rate and not by limiting the capacity as pointed out by van Laarhoven (2017). The two paragraphs below (12) are contradictory or I'm missing something: first paragraph says that "This is contrary to our intuition that weight decay results in a simple function." but immediately below, "We show empirically that weight decay only improves generalization by controlling the norm, and therefore the effective learning rate." Can the authors please explain what the "effective learning rate" argument is?

Proposition 1 and theorem 1 are extensions from Martens & Gross, 2015, I didn't fully check the calculations. I glanced through them, and they mostly use algebraic manipulations. The main empirical takeaway as the authors mention is that: weight decay in both KFAC-F and KFAC-G serves as a complexity regularizer which sounds trivial (assuming Martens & Grosse, 2015) since in both of these cases, BN is not used and the fact that weight decay is regularization using the local norm.

If I understand correctly, KFAC is an approximate second order method with the approximation chosen to be such that it is invariant under affine transformations. Are there any convergence guarantees at all for either of these approaches? Newton's method, even for strongly convex loss functions, requires self-concordance to ensure convergence, so I'm a bit skeptic when using approximate (stochastic) Jacobian norm.

Some of the plots have loss values, some have accuracy etc., which is also confusing while reading. I strongly suggest that Figure 1 be shown differently, especially the x-axis! Essentially weight decay improves the accuracy about 2-4% but it is hard to interpret that from the figure.

---

> ### Author Response · Authors · 2018-11-06
> **Response to AnonReviewer2**
>
> Thank you for the insightful comments. According to your suggestions, we revised the statements of the paper (including 4.1) to make them clearer.
>
> Q1: what is the definition of "effective learning rate"
> For "effective learning rate", you can understand it as the "learning rate" for normalized networks (see equation 9).
>
> Q2: regularization just changes the learning rate (Mechanism 1)
> Note: it's true for weight decay in general (not L2-norm). We also tested weight decay in the case of Adam (see Figure 2) where weight decay and L2 regularization are not identical.
>
> Q3: why reducing the scale of the weights increase the effective learning rate
> As explained in equation 9, the effective learning is inversely proportional to the weight norm.
>
> Q4: The sentence starting (in point #1.) with "As evidence,", what is the evidence for?
> See Figure 2 and Figure 4. Most of the generalization effect of weight decay is due to applying it to layers with BN.
>
> Q5: The improvement provided by weight decay is uniform across the board.
> Weight decay does improve the performance consistently, but the mechanisms behind are different (depending on the optimization algorithm and network architecture). Figure 1 and Table 1 are mostly to emphasize the difference between L2 regularization and weight decay so as to motivate three mechanisms.
>
> Q6: Argument of Mechanism 1 (or effective learning rate)
> In mechanism 1, we basically argue that the scaling of weights for BN layers doesn't influence the underlying function (see equation 8), so it doesn't meaningfully constrain the function to be simple (you can always scale down the weights but the function represented by the network is still the same, also see the first paragraph in 4.1). However, the scaling of the weights does influence the updates (see equation 9) by controlling the effective learning rate. The regularization effect of weight decay is achieved by scaling the weights, and therefore the effective learning rate.
>
> Q7: Proposition 1 and Theorem 1 are extensions from Martens & Gross, 2015
> We have removed Proposition 1 in the latest version. Theorem 1 (Lemma 2 in the latest version) is not just an extension from the K-FAC paper (martens & Grosse, 2015). Actually, it has little to do with the K-FAC paper.  We don't think it's trivial for the following reasons:
>
> - Theorem 1 (Lemma 2 in the latest version) is new and it heavily relies on the Lemma 1 (gradient structure) which has nothing to do with the original K-FAC (Martens & Grosse, 2015) paper.
> - Theorem 1 (Lemma 2 in the latest version) is an important part to connect Gauss-Newton norm to approximate Jacobian norm. The result of approximate Jacobian norm is non-trivial and we didn't see any similar theoretical result before. In practice, it's quite expensive to directly regularize Jacobian norm due to the extra computation overhead. In this work, we provide a simple yet cheap way to approximately regularize Jacobian norm and we believe it's useful and novel.
>
> Q8: K-FAC (convergence?)
> K-FAC is currently the most popular approximate natural gradient method in training deep neural networks. It works very well (due to the use of curvature information) in practice and we didn't see any convergence issue. Recently, Bernacchia, 2018 [1] provided convergence guarantee for natural gradient in the case of deep linear networks (where the loss is non-convex). Beyond that, they also gave some theoretical justifications for the performance of K-FAC.
>
> [Reference]
> [1] Exact natural gradient in deep linear networks and application to the nonlinear case

---

### Author Response · Authors · 2018-11-06
**Paper revision**

We have updated the paper and improved the writing a lot. In particular, we rewrote the section 4.1 and 4.2 as requested by AnonReview1 and AnonReview2.

---

### Public Comment · (anonymous) · 2018-11-15
**Related work**

The first mechanism, increasing the effective learning rate, is also identified in this work https://arxiv.org/abs/1709.04546 (Sec. 2.2 and 3.2). The authors may want to discuss how they are related.

---

> ### Author Response · Authors · 2018-11-15
> **Re: Related work**
>
> Thank you for pointing out this work.
>
> Section 2.2 of this paper is indeed related to our mechanism 1. However, the argument of effective learning rate was first identified by van Laarhoven 2017 and we did properly discuss the relationship with van Laarhoven 2017 (also see the response to AnonReview1). In the upcoming version, we will cite the paper you mentioned.

---

### Meta-Review · Area_Chair1 · 2018-12-14

**Confidence:** 4
**Recommendation:** Accept (Poster)

**Metareview:**

Reviewers are in a consensus and recommended to accept after engaging with the authors. Please take reviewers' comments into consideration to improve your submission for the camera ready.